First characterization of the probiotic potential of lactic acid bacteria isolated from Costa Rican pineapple silages

Wen Fang Wu Wu Jannette 1
Redondo-Solano Mauricio 2
Uribe Lidieth 3
WingChing-Jones Rodolfo 4
Usaga Jessie 5
Barboza Natalia natalia.barboza@ucr.ac.cr 6
1 Food Technology Department, Universidad de Costa Rica , San Pedro , San José , Costa Rica
2 Research Center for Tropical Diseases (CIET) and Food Microbiology Research and Training Laboratory (LIMA), College of Microbiology, University of Costa Rica (UCR), Universidad de Costa Rica , San Pedro , San José , Costa Rica
3 Agronomic Research Center (CIA), Universidad de Costa Rica , San Pedro , San José , Costa Rica
4 Animal Science Department, Animal Nutrition Research Center (CINA), Universidad de Costa Rica , San Pedro , San José , Costa Rica
5 National Center for Food Science and Technology (CITA), Universidad de Costa Rica , San Pedro , San José , Costa Rica
6 Food Technology Department, National Center for Food Science and Technology (CITA), Center for Research in Cellular and Molecular Biology (CIBCM), Universidad de Costa Rica , San Pedro , San José , Costa Rica
Collado Maria Carmen
Electronic publication date: 2021 Nov 30
Publication date: 2021
Volume: 9
Electronic Location ID: e12437
Received 2021 May 13; Accepted 2021 Oct 15
Copyright: ©2021 Wen Fang Wu Wu et al.
Copyright year: 2021
Copyright holder: Wen Fang Wu Wu et al.
License: This is an open access article distributed under the terms of the Creative Commons Attribution License, which permits unrestricted use, distribution, reproduction and adaptation in any medium and for any purpose provided that it is properly attributed. For attribution, the original author(s), title, publication source (PeerJ) and either DOI or URL of the article must be cited.
License URL: https://creativecommons.org/licenses/by/4.0/

Keywords: Agro-industrial residuals, Gastrointestinal tract survivor, Pathogens, Inhibition, Antibiotics resistance

Funding: The Costa Rican Ministry of Science and Technology (MICITT) The University of Costa Rica (UCR), Projects FI-031B-14 B9-457 B8-610 This study was supported by the Costa Rican Ministry of Science and Technology (MICITT) and the University of Costa Rica (UCR), Projects FI-031B-14, B9-457 and B8-610. The funders had no role in study design, data collection and analysis, decision to publish, or preparation of the manuscript.

==============================
Background

Agro-industrial waste from tropical environments could be an important source of lactic acid bacteria (LAB) with probiotic potential.

Methods

Twelve LAB isolates were isolated from pineapple silages. The species identification was carried out considering 16S rRNA and pheS genes. Experiments to evaluate the probiotic potential of the isolates included survival under simulated gastrointestinal environment, in vitro antagonistic activity (against Salmonella spp. and Listeria monocytogenes), auto-aggregation assays, antibiotic susceptibility, presence of plasmids, adhesiveness to epithelial cells, and antagonistic activity against Salmonella in HeLa cells.

Results

Lacticaseibacillus paracasei, Lentilactobacillus parafarraginis, Limosilactobacillus fermentum, and Weissella ghanensis were identified. Survival of one of the isolates was 90% or higher after exposure to acidic conditions (pH: 2), six isolates showed at least 61% survival after exposure to bile salts. The three most promising isolates, based on survivability tests, showed a strong antagonistic effect against Salmonella. However, only L. paracasei_6714 showed a strong Listeria inhibition pattern; this isolate showed a good auto-aggregation ability, was resistant to some of the tested antibiotics but was not found to harbor plasmids; it also showed a high capacity for adhesion to epithelial cells and prevented the invasion of Salmonella in HeLa cells. After further in vivo evaluations, L. paracasei_6714 may be considered a probiotic candidate for food industry applications and may have promising performance in acidic products due to its origin.

Introduction

Currently, the development and intake of functional foods containing probiotic microorganisms have grown considerably due to their known health benefits and ability to prevent certain diseases (Nami et al., 2018). Probiotics are defined by the Food and Agriculture Organization of the United Nations and the World Health Organization (FAO/WHO) as “microorganisms which when administered in adequate amounts confer a health benefit on the host” (FAO/WHO, 2002). Probiotics are capable of enduring gastrointestinal (GI) tract conditions, to temporarily colonize the intestinal environment and supply health effects through modulation of GI microbiota and immunogenic responses, or by producing certain beneficial metabolites of interest (Meybodi & Mortazavian, 2017; Nami et al., 2018). Delivery of health-promoting microorganisms is commonly done through the consumption of fermented products, most frequently dairy (Nascimento et al., 2019). However, with the increased incidence of lactose intolerance, vegetarianism, and other consumer demands, interest in the development of non-dairy probiotic foods has grown. Nevertheless, changes in matrix properties may imply variations in the probiotic physiological dynamics (Dey, 2018).

The majority of probiotic bacteria belong to the lactic acid bacteria (LAB) group that are capable to produce antimicrobial compounds such as lactic acid and bacteriocins (Soccol et al., 2010), which makes them suitable as probiotics and bio-control organisms due to their ability to inhibit other microorganisms through the production of different metabolites or by competitive exclusion (Vieco-Saiz et al., 2019).

The genera Lactobacillus and Bifidobacterium are commonly used probiotics. However, Lactococcus, Streptococcus, Enterococcus, and selected yeasts can potentially be used as probiotics as well (De Vrese & Offick, 2010; Ayala et al., 2019). The selection and characterization of novel microorganisms as potential probiotics must take into account certain properties such as tolerance to low pH and high bile salt concentrations, as these conditions are present in the GI tract environment during digestion processes (García-Ruiz et al., 2014; Byakika et al., 2019). Recent studies have also suggested the importance of evaluating other features such as adhesiveness to the intestinal mucosa, prolonged and stable persistence in the GI tract, and antimicrobial properties (García-Ruiz et al., 2014).

In the last years, probiotics have been obtained mostly from fermented dairy products or the human GI tract (Kook et al., 2019). Nonetheless, with the increasing demand for novel probiotics with improved health and processing properties, the search for organisms from non-traditional sources has been intensified (Kumar et al., 2015). Some of the unconventional sources that have recently been screened for potential probiotics include traditional fermented foods and beverages, vegetables, and vegetable wastes (Sornplang & Piyadeatsoontorn, 2016; Ruiz-Rodríguez et al., 2019). Different intrinsic characteristics of these matrices are considered significant factors leading to the diversity of species or isolates that can be found (Sornplang & Piyadeatsoontorn, 2016). In fact, LAB isolated from non-traditional foods can show better performance and high competitiveness as food additives (Somashekaraiah et al., 2019).

Multiple sources to isolate LAB with probiotic potential can be found in tropical and subtropical environments. In the Latin-American region, different research have been carried out in terms of screening and evaluation of new LAB isolates with health-promoting properties. Most of the studies have focused on the isolation of strains from local foods (Maldonado et al., 2012; Melgar-Lalanne et al., 2013; Ramos et al., 2013; Agostini et al., 2018), food animals (Iñiguez Palomares, Pérez-Morales & Acedo-Félix, 2007), and traditional beverages (Romero-Luna, Hernández-Sánchez & Dávila-Ortiz, 2017). A minor portion of the studies has evaluated strains obtained from environmental sources such as fruits (Veron et al., 2017), rain forest (Benavides et al., 2016), and agro-industrial products (Schwan, 1998; Santos et al., 2016). However, the characterization of LAB with probiotic potential has not been performed in Costa Rica yet.

The aim of this research was to assess the probiotic potential of autochthonous LAB isolated from Costa Rican pineapple peel silages. Selected LAB isolates were identified using molecular markers and subjected to a series of in vitro analyses to evaluate (a) resistance to GI tract conditions; (b) antimicrobial properties, (c) auto-aggregation ability, (d) safety properties, and (e) adhesion to epithelial cells. These evaluations were done as a preliminary screening for strains with potential application in fermented food applications. This is the first report of the evaluation of LAB with promissory probiotic traits from silages of pineapple residuals from Costa Rica.

Materials & Methods

Isolation of bacterial isolates

Lactic acid bacteria were isolated from twenty pineapple peel samples that were vacuum-ensiled for 30 days. The samples were obtained from a Costa Rican company dedicated to pineapple juice production (WingChing-Jones et al., 2021). Twenty-five grams of each sample was homogenized with 0.1% w/v peptone water (PW) (Oxoid, Basingstoke, UK) and serially diluted in tubes containing 9 mL of deionized water. Each dilution was used to streak De Man, Rogosa, and Sharpe agar plates (MRS) (Difco, Le Pont de Claix, France) that were incubated at 35 ± 2 °C overnight in anaerobic conditions. Selected colonies were subjected to Gram staining and a posterior morphological identification. The cultures were stored as glycerol stocks (20% v/v) at −80 °C until analyzed. All accessions are kept (with the same name indicated on this research) in the Bacteriology Collection at the Faculty of Microbiology and in the Bacteriology Collection at the National Center for Food Science and Technology (CITA), University of Costa Rica. The strain L. casei ATCC 393 was used as a control given that it is currently commercialized as probiotic (Sidira et al., 2010; Haddaji et al., 2015).

DNA extraction and PCR amplification

Total nucleic acids were extracted from each isolate using a miniprep protocol (Birnboim & Doly, 1979). A 1.5 kb fragment of the 16S rRNA gene was amplified using the primer pair 27F/1492R (Edwards et al., 1989). The PCR was done considering the conditions of an initial denaturation step at 94 °C for 1 min, 30 cycles of 94 °C for 40 s, 55 °C for 1 min, 72 °C for 1 min, and a final extension at 72 °C for 5 min. The master mix contained a final volume of 25 µl and included 1X reaction buffer, 0.2 mM dNTPs, 0.2 µM of each primer, 1.5 mM MgCl2, 1 U Taq DNA polymerase (Bio-Rad, Hercules, CA, USA), and 50 ng of DNA. In addition, a ∼490 bp fragment of the phenylalanyl-tRNA synthase (pheS) gene was amplified by PCR using the primer pair combination pheS-21-F/pheS-22-R (Naser et al., 2005). The reaction was performed using iProof High-Fidelity DNA polymerase (Bio-Rad) and 50 ng of DNA. The following cycling conditions were used: 98 °C for 30 s, 35 cycles of 98 °C for 30 s, 60 °C for 30 s, and 72 °C for 30 s; and a final extension at 72 °C for 10 min. PCR products were visualized by electrophoresis in a 1% agarose gel and stained with GelRed (10.000 X) (Biotium, Fremont, CA, USA). The amplified gene fragments were sequenced in both orientations by Macrogen® (Seoul, South Korea).

Sequencing analysis

The Staden package was used to assemble the obtained sequences. Sequences were aligned using the MUSCLE algorithm (MEGA 7) (Kumar, Stecher & Tamura, 2016). Sequences were compared with those available in the databases with the BlastN tool (Altschul et al., 1990). Costa Rican sequences were deposited in the GenBank (Table S1). A total of 25 LAB sequences (12 isolates from this study and 13 obtained from GenBank) were used for phylogenetic comparison. A region of 1,299 nucleotides (nt) corresponding to 16S rRNA gene and a fragment of 420 nt for the pheS gene, were selected. A phylogenetic tree was constructed using Bayesian phylogenetic analysis. Ten million generations, eight chains, and a mixed model with sampling every 1.000 generations was considered (Huelsenbeck & Ronquist, 2001; Ronquist & Huelsenbeck, 2003). As an external group, the sequences of L. delbrueckeii subsp. lactis KTCT 3034 was considered for phylogenetic analysis of both genes. Sequences obtained on this research are shown in bold font.

Assays of resistance to the gastrointestinal tract

Tolerance to pH 2.0. All isolates and a control strain (L. casei ATCC 393) were exposed to pH 2.0 (Ramos et al., 2013), in order to evaluate tolerance to acidic conditions. Each isolate was cultivated in MRS broth (Difco) at 35 ± 2 °C for 24 h and pH 7.0. Cells were centrifuged at 5,000 rpm for 5 min at 24 °C, washed two times in PW (Oxoid), and resuspended in PW (Oxoid) to a concentration of about 108 CFU/mL. A 1 mL aliquot of the final bacterial suspension was used to inoculate 50 mL of MRS broth (Difco) adjusted to pH 2.0 using 1 N HCl (Thermo Fisher Scientific, Waltham, Massachusetts, USA) and cultures were incubated at 35 ± 2 °C for 3 h. After 3 h of incubation, the effect of acidity was neutralized with 1N NaOH (Thermo Fisher Scientific, Waltham, Massachusetts, USA). To quantify the final bacterial population, 1 mL aliquots obtained at time 0 and after 3 h incubation were serially diluted in PW (Oxoid), plated on MRS agar (Difco), and incubated in anaerobic jars for 72 h at 35 ± 2 °C. The assay was conducted in triplicate.

Lysozyme resistance

Lysozyme resistance was evaluated using a modified version of the method described by Zago et al. (2011). One milliliter of LAB cells and a control strain (L. casei ATCC 393) was cultured in MRS broth (Difco) at 30 ± 2 °C for 24 h. After incubation, an aliquot of the culture was centrifuged at 5,000 rpm for 5 min at 24 °C and washed twice in phosphate buffer (0.1 M, water pH 7.0). The bottom was resuspended in two mL of Ringer solution (8.5 g/L NaCl, 0.4 g/L KCl, 0.34 g/L hydrated CaCl2) (Sigma Aldrich, St. Louis, MO, USA). A sterile electrolyte solution (SES) (0.22 g/L CaCl2, 6.2 g/L NaCl, 2.2 g/L KCl, 1.2 g/L NaHCO3) containing 100 mg/L of lysozyme (Sigma Aldrich) was used to resuspend each LAB (108 CFU/mL). Bacterial suspensions in SES without lysozyme were used as negative controls. Each sample was incubated in a water bath at 37 °C for 0, 30, and 120 min. After incubation, serial dilutions were made in PW (Oxoid) and samples were plated in duplicate on MRS and incubated for 72 h at 35 °C under anaerobic conditions. Cell counts were done, and survival was determined according to the population described as the percentage of CFU/mL after 30 and 120 min relative to the bacterial population in CFU/mL at time zero. Assays were carried out in triplicate.

Resistance to bile salts

LAB tolerance to bile salts was evaluated following the protocol described by García-Ruiz et al. (2014)) with minor modifications. The isolates that showed a survival greater than 20% after exposure to pH 2 and lysozyme were selected. The isolates were grown overnight in MRS (Difco) and independently inoculated (2% v/v) in fresh MRS broth (Difco) supplemented with 0.3% bile salt (w/v) (Sigma-Aldrich). The LAB was incubated in tilted tubes at 35 ± 2 °C for 24 h and shacked at 250 rpm in a rotary benchtop incubated shaker (Lab Companion model SI-600R, Jeio Tech Company, South Korea). Counts were performed following the procedure previously described. A sample without bile salts was used as a control. Every experimental trial was performed in triplicate and the growth percentage of each culture was compared to the control.

Antimicrobial assays

Antagonistic activity against pathogens

The antagonistic activity of all isolated LAB isolates and L. casei ATCC 393 against Listeria monocytogenes and Salmonella enterica was evaluated using a modified version of the overlay protocols (Booth, Johnson & Wilkins, 1977; Hütt et al., 2006; Soleimani et al., 2010). Five L. monocytogenes strains were used, including four isolates from processed meat products and one reference strain (ATCC 19116). The five Salmonella isolates used in the study included one Salmonella serovar Typhimurium, one S. Typhi, and three isolates of undefined serotype. Before the experiments, each LAB and pathogen strain was individually grown at 35.0 ± 0.5 °C for 24 ± 2 h in MRS (Difco) or Tryptic Soy Broth (TSB) (Oxoid), respectively. After incubation, each LAB was inoculated on MRS agar plates in a thick straight line approximately 7 cm in length and 0.5 cm from the edge; streaked plates were incubated under capnophilic conditions at 35.0 ± 0.5 °C for 24 ± 2 h. The MRS plates were then overlaid with approximately 5 ml of Brain Heart Infusion agar (BHI) (Oxoid). After solidification, plates were swabbed with a cocktail suspension prepared with the overnight cultures of each pathogen. Petri dishes were incubated at 35.0 ± 0.5 °C for 24 ± 2 h under aerobic conditions. The plates were then examined for a clear inhibition zone around the line of each LAB. Clear zones were measured, and inhibitory activity was determined (Pan et al., 2009). Inhibition zones with a diameter larger than 6 mm were considered a confirmation of strong antagonistic activity.

Antimicrobial activity of the supernatants

The antimicrobial activity of the cell-free supernatants was determined against the same pathogenic strains by using a previously described protocol with modifications (Lourenço & Pinto, 2011). The isolate L. paracasei_6714, which showed inhibition zones with a diameter larger than 6 mm for both pathogens, was cultured in MRS broth (Oxoid) at 35 ± 0.5 °C for 24 ± 2 h. The LAB cultures were centrifuged at 1,500 rpm for 15 min and the supernatant was decanted and filtered (0.2 µm) into sterile test tubes. To avoid an inhibitory effect due to acid lactic exposure, the pH of the supernatant was adjusted to 7.00 with a solution of 0.1 M NaOH (Thermo Fisher Scientific, Waltham, MA, USA) and the supernatant was used immediately. An isolated colony of each pathogenic strain grown overnight on Tryptic Soy Agar (TSA) (Oxoid) was suspended in PW (Oxoid) to obtain a McFarland standard of 0.5; equal volumes of each strain suspension were mixed to obtain the cocktail solutions used in the experiments. The wells of a 96-well microplate were filled with a 50 µL of sterile TSB (Oxoid), 50 µL of the indicator pathogen solution, and variable volumes (50, 45, 40, 35, 30, 25, 20, and 15 µL) of filtered supernatant adjusted to 50 µL with sterile MRS (Difco). Positive and negative controls were included. The positive control was prepared with 50 µL of sterile TSB (Oxoid), 50 µL of the indicator pathogen, and 50 µL of sterile MRS (Difco). Negative controls did not contain the pathogen, and the volume was adjusted with 50 µL of sterile PW (Oxoid). Microplates were incubated aerobically at 35.0 ± 0.5 °C for 24 ± 2 h in high humidity conditions and the absorbance at 620 nm was measured in an Ultra Microplate Reader (Biotek Instruments, Winooski, VT, USA). Results were adjusted by subtracting the absorbance value obtained for the negative control. All determinations were performed in triplicate. To analyze the inhibitory effect of the supernatant solutions on the two pathogens, two-way analysis of variance (ANOVA) followed by Tukey’s honest significant difference test were performed using JMP version 11 (SAS Institute Inc., USA). Differences were considered significant at a P-value of < 0.05.

Auto-aggregation assays

The auto-aggregation assay was performed following the protocol described by Rastogi, Mittal & Singh (2020) with some modifications. L. paracasei_6714, L. fermentum_6702 and L. casei ATCC 393 (control) were grown in MRS broth at 35 ± 2 °C for 24 h and were later harvested through centrifugation (10.000×g for 15 min, 4 °C), washed twice with phosphate buffer solution (PBS) (50 mM KH2PO4/K2HPO4, pH 6.8) (Sigma-Aldrich, San Luis, Missouri, USA) and resuspended in PBS to obtain an absorbance of around 0.8 at 600 nm. three mL of bacterial suspension was vortexed and incubated at room temperature for 4 h. Every hour, 0.1 mL of upper suspension was transferred to 3.9 mL of PBS and the OD600 was measured. PBS was used as blank.

The auto-aggregation percentage was then calculated using the equation: Ao−AtAo∗100=%CellularAuto−aggregation

where At is the OD600 at time t (t = 1, 2, 3, 4) and Ao is the OD600 at t = 0.

Safety assays

Antibiotic resistance

The antibiotic sensitivity of isolate L. paracasei_6714 was evaluated by following the swab and agar disk diffusion method (Hudzicki, 2009). A complete set of antibiotics comprising different families was used. The LAB isolate was cultured in MRS broth (Oxoid) at 35 ± 0.5 °C for 24 ± 2 h and the suspension of the test isolate was swabbed on solidified Müller-Hinton agar (Oxoid) using a sterile cotton swab. Antibiotic disks impregnated with ciprofloxacin (5 µg), vancomycin (30 µg), penicillin (10 IU), amoxycillin with clavulanic acid (30 µg), erythromycin (15 µg), amikacin (30 µg), streptomycin (10 µg), tetracycline (30 µg) and chloramphenicol (30 µg) (Liofilmchem, Vie a Scozia, Italy) were placed on the agar plates. Plates were incubated at 35 ± 0.5 °C for 24 ± 2 in capnophilic conditions. After incubation, the diameter of the inhibition zones was measured and compared with the standards established by the Clinical and Laboratory Standard Institute (Sharma et al., 2016; Wolupeck et al., 2017). Experimental trials were performed in triplicate.

Plasmid DNA isolation

L. paracasei_6714 was cultured in MRS broth (Oxoid) at 35 ± 0.5 °C for 24 ± 2 h. Plasmid DNA was extracted using a QIAprep Spin Miniprep Kit (Qiagen, Hilde, Germany). The DNA was run and visualized in a 0.8% agarose gel stained with GelRed® (Biotium, Fremont, CA, USA). Plasmid size was estimated using a using a 100 bp MassRuler DNA ladder (Thermo Fisher Scientific).

Cell culture assays

Preparation of cell monolayer

The in-vitro adhesion of L. paracasei_6714 was assayed using HeLa cells (kindly supplied by the Research Center for Tropical Diseases, University of Costa Rica). Cells were cultured in a monolayer of Eagle’s Minimum Essential Media (EMEM) (Thermo Fisher Scientific) supplemented with 10% v/v fetal bovine serum, 20 µM glutamine per mL, 50 U penicillin G, and 50 µg/mL of streptomycin. Cultured cells were incubated at 35 ± 0.5 °C in a modified atmosphere of 5% CO2 and 95% O2 until used. Before experiments were conducted, the EMEM (Thermo Fisher Scientific) was discarded and cells were washed with 5 mL of 10X PBS (Sigma-Aldrich). Cells were then covered with a solution of 2.5 mL of trypsin and EDTA 0.05 (GIBCO, Thermo Fisher Scientific) with phenol red (GIBCO, Thermo Fisher Scientific) and incubated for 3 min to promote cell separation. Detached cells were resuspended in 2.5 ml of EMEM (Thermo Fisher Scientific), and a small volume was obtained for cell quantification using a Neubauer chamber. A 12-well microplate was filled with different volumes of cell suspensions and 2 mL of EMEM (Thermo Fisher Scientific) to obtain a cell concentration of 106 cells/ml and then incubated for 48 h, as previously indicated.

In-vitro cell adhesion assay

A modified version of a previously published methodology was used (Gopal et al., 2001; Tsai et al., 2005). L. paracasei_6714, at a concentration of about 107 CFU/mL in EMEM (Thermo Fisher Scientific), was placed over a monolayer of HeLa cells previously grown on a glass slide incubated inside a 12-well microplate. Microplates were then incubated for 2 h at 35 ± 0.5 °C. After incubation, cells were washed twice with PBS (Sigma-Aldrich), fixed with 10% of paraformaldehyde for 10 min, washed twice with PBS (Sigma-Aldrich), and then stained with crystal violet for 5 min. The stained slides were washed with PBS (Sigma-Aldrich) to remove the excess dye and observed under a light microscope. LAB adhesion was evaluated by quantifying the mean number of bacterial cells attached to the HeLa cell monolayer in 5 randomly selected microscopic fields. L. paracasei counts were determined for an average of 26 epithelial cells. A positive control with L. fermentum_6702 (low adhesion capacity isolate determined in preliminary assays not included here) was included for comparison.

Antagonistic effect of L. paracasei against Salmonella invasion in HeLa cells

Treatment assay

A modified version of a previous published methodology was used (Giannella et al., 1973). Salmonella serovar Typhimurium was grown on TSB (Oxoid) at 35 ± 0.5 °C for 24 ± 2 h and diluted in antibiotic-free EMEM to obtain a concentration of about 107 CFU/mL. L. paracasei_6714 was grown in MRS (Oxoid) incubated under the same conditions and then diluted as described for Salmonella. A volume of 1 mL of each culture suspension was added to each cell monolayer inside the 12-well microplate. Plates were centrifuged at 1,600 rpm for 5 min and then incubated for 0, 3, and 24 h under the same conditions described for cell maintenance. After incubation, wells were washed two times with PBS and then kept for 1 h in fresh EMEM (Thermo Fisher Scientific) medium containing 100 µg/mL of gentamicin. After gentamicin exposure, each well was washed twice with PBS (Sigma-Aldrich) and cells were then lysed with ultrapure water for 10 min. Appropriate dilutions in PW (Oxoid) were spread onto TSA (Oxoid) and xylose lysine deoxycholate agar (XLD) (Oxoid). The plates were incubated at 35 ± 0.5 °C overnight. Bacterial counts were used to calculate the invasion rate. A positive control of Salmonella was included. Experiments were performed in triplicate.

Protection assay

The protocol described for the treatment assay was modified to include pre-exposure of each cell monolayer to L. paracasei_6714 for 3 and 24 h before infection with Salmonella.

Results

A total of twelve different LAB morphotypes were isolated from twenty pineapple silages with increasing levels of urea. Considering the 16S rRNA sequence and pheS gen the isolates correspond to L. paracasei (seven isolates), Lentilactobacillus parafarraginis (two isolates), Limosilactobacillus fermentum (two isolates), and W. ghanensis (one isolate) (Table 1 and Table S1). When the sequences obtained in this research and those selected from GenBank (https://www.ncbi.nlm.nih.gov/genbank/) were considered, a clear cluster was established (Fig. 1). Equivalent length portions of both genes were used to resolve the species groups obtained. The species were renamed according to the novel classification of Zheng et al. (2020). Isolates of L. paracasei were also previously characterized with multilocus typing sequences (MTLS). Results were reported by WingChing-Jones et al. (2021).

Table 1 Sequence of primers used for identification of lactic acid bacteria (LAB) from this research.

Primer name	Forward primer (5′→ 3′)	Reverse primer (5′→ 3′)	Location a	
27F/1492R	AGA GTT TGA TCC TGG CTC AG	ACG GCT ACC TTG TTA CGA CTT	259 513…261 026	
pheS-21-F/pheS-22-R	CAYCCNGCHCGYGAYATGC	CCWARVCCRAARGCAAARCC	1 670 081…1 670 575	
Notes.

a Location on the genome of strain L. paracasei ATCC 334 (GenBank accession no. CP000423) of the primers.

Figure 1 Phylogeny based on Bayesian analysis and considering the partial sequences of the 16S rRNA gene (1,299 nucleotides (nt)) (A) and phenylalanyl-tRNA synthase gene (pheS) (420 nt) (B) of lactic acid bacteria (LAB) isolated from ensiled pineapple peels.

Probabilities are indicated at nodes. As an external group. L. delbrueckeii subsp. lactis KTCT 3034 was used as an external sequence for both figures. Sequences obtained on this research are shown in bold font.

After exposure to acidic conditions (pH 2.0), all LAB isolates were viable, but just one isolate (L. parafarraginis 6719) showed a population that survived more than 90%. No reduction was observed in the population of the control samples (pH 6.0) as expected (Table 2) and total reduction was observed in the case of the control isolate L. paracasei ATCC 393. A higher rate of survival was also observed for L. paracasei (isolates: 6710 and 6715) with values of 52.6% and 42.9%, and L. fermentum (isolates: 6702 and 6704) with values of 31.2% and 22.1%, respectively. On the other hand, eight isolates showed more than 90% of survival after 30 min exposure to lysozyme but just six of them were able to fulfill these criteria after 120 min of exposure. Among those isolates showing higher resistance to low pH, just isolates 6704 and 6710 had a survivability of more than 90% to lysozyme after 120 min of exposure. Interestingly, L. parafarraginis 6719 was very sensitive to the effect of lysozyme (13.1% of survival after 120 min). Given that any of the LAB isolates fulfilled the selection criteria, isolates showing higher resistance to both conditions (pH and lysozyme) were selected for the bile tolerance test. Survival was lower than 10% in all the cases, but higher resistance was observed for L. parafarraginis 6719 (8.8%) and L. fermentum 6702 (2.1%). Still, tolerance to bile salts was lower for the control strain (L. casei ATCC 393), a commercially available probiotic culture, when compared with the other isolates.

The antagonistic activity of the twelve isolates and the control (L. casei ATCC 393) from this study against selected pathogens is shown in Table 3 and Fig. S1. Three isolates produced strong inhibition zones against Salmonella. Nevertheless, when the isolates were evaluated against L. monocytogenes, only one isolate (L. paracasei_6714) produced an inhibition zone with a diameter greater than the reference criteria (6 mm). According to these results, the antimicrobial activity of the supernatant of L. paracasei_6714 was evaluated and the results are shown in Table 4. Significant inhibition of Salmonella was observed with 20 µL of the supernatant, while up to 50 µL were required to obtain the same effect for Listeria.

The auto-aggregation ability of L. paracasei_6714, L. fermentum_6702, and L. casei ATCC 393 (control) was measured at four consecutive time intervals (1, 2, 3, and 4 h). The results conveyed in Fig. 2 in which is shown a steady increase in auto-aggregation by the studied isolates. After 4 h, L. fermentum_6702 showed the lowest auto-aggregation percentage, while L. casei ATCC 393 and L. paracasei_6714 presented a good auto-aggregation ability, suggesting an effective cell adhesion capacity.

The antibiotic susceptibility of L. paracasei_6714 is shown in Table 5. The isolate was resistant to most of the tested compounds. The only exceptions were amoxicillin with clavulanic acid and erythromycin, where an intermediate sensitivity was observed. In addition, the L. paracasei_6714 isolate isolated was not found to harbor plasmids, which indicates a low probability of transferring the antibiotic resistance feature (Fig. S2).

Table 2 Resistance/tolerance to pH 2.0, lysozyme and bile salts of lactic acid bacteria (LAB) isolated from pineapple silage.

LAB strain	Tolerance to pH 2.0	Resistance to lysozyme			Resistant to bile at 0.3 %	
							t 30	t 120					
	Control (log CFU/ml)	Initial population (log CFU/ml)	Final population (log CFU/ml)	Survival (%)	Control (log CFU/ml)	Initial population (log CFU/ml )	Final population (log CFU/ml)	Survival (%)	Final population (log CFU/ml)	Survival (%)	Control (log CFU/ml)	Initial population (log CFU/ml)	Final population (log CFU/ml)	Survival (%)	
L. casei ATCC 393 (control)	8.50	8.1 ± 1.7	0.00 ± 0	<90%	8.25	7.39 ± 0.09	7.59 ± 0.18	100%	8.45 ± 0.06	100%	8.95	9.0 ± 1.2	3.8 ± 1.1	<50%	
L. paracasei_6709	6.83	6.56 ± 0.06	2.94 ± 0.02	<90%	8.37	8.26 ± 0.18	8.37 ± 0.10	100%	8.21 ± 0.14	90.31 ± 10.7	ND	ND	ND	ND	
L. paracasei _6710	6.99	6.57 ± 0.09	6.28 ± 0.05	<90%	7.96	7.93 ± 0.17	7.95 ± 0.14	100%	7.95 ± 0.16	100%	9.61	9.2 ± 0.5	5.5 ± 0.4	<50%	
L. paracasei _6711	7.70	7.6 ± 0.7	6.49 ± 0.06	<90%	7.97	7.96 ± 0.16	8.02 ± 0.16	100%	7.86 ± 0.16	<90%	ND	ND	ND	ND	
L. paracasei _6712	6.83	6.79 ± 0.01	5.71 ± 0.02	<90%	8.45	8.16 ± 0.05	8.27 ± 0.07	100%	8.49 ± 0.35	100%	ND	ND	ND	ND	
L. paracasei _6713	6.18	5.99 ± 0.004	5.27 ± 0.01	<90%	8.02	8.12 ± 0.15	8.15 ± 0.16	100%	8.03 ± 0.11	<90%	ND	ND	ND	ND	
L. paracasei _6714	5.92	5.69 ± 0.05	4.55 ± 0.07	<90%	8.13	8.40 ± 0.25	8.23 ± 0.06	<90%	8.30 ± 0.15	100%	ND	ND	ND	ND	
L. paracasei _6715	7.04	5.98 ± 0.07	5.6 ± 0.1	<90%	7.72	8.27 ± 0.28	8.08 ± 0.13	<90%	7.93 ± 0.04	<90%	9.76	9.6 ± 0.2	7.1 ± 0.5	<50%	
L. fermentum_ 6702	6.99	6.48 ± 0.02	5.97 ± 0.03	<90%	8.51	8.41 ± 0.06	8.48 ± 0.32	100%	8.29 ± 0.14	<90%	8.30	8.3 ± 0.1	6.5 ± 0.5	<50%	
L. fermentum_6704	6.90	6.59 ± 0.02	5.93 ± 0.04	<90%	8.50	8.36 ± 0.16	8.35 ± 0.20	97.5 ± 10.0	8.41 ± 0.17	100%	10.23	9.5 ± 0.6	7.5 ± 0.5	<50%	
L. parafarraginis_ 6717	6.79	6.67 ± 0.01	5.766 ± 0.004	<90%	8.16	8.50 ± 0.01	6.57 ± 0.02	<90%	6.44 ± 0.01	<90%	8.91	ND	ND	ND	
L. parafarraginis_ 6719	7.70	7.64 ± 0.01	7.62 ± 0.01	95.4 ± 2.3	8.00	7.82 ± 0.15	7.59 ± 0.16	<90%	6.93 ± 0.11	<90%	9.08	9.04 ± 0.04	8.00 ± 0.1	<50%	
W. ghanensis_ 6706	5.48	5.64 ± 0.06	4.4 ± 0.1	<90%	6.30	6.88 ± 0.18	6.19 ± 0.24	<90%	6.18 ± 0.03	<90%	ND	ND	ND	ND	
Notes.

ND not determined

Mean values (± standard deviation, n = 3).

Table 3 Inhibition halo of Salmonella enterica and Listeria monocytogenes grown on culture media pre-inoculated with different LAB strains isolated from pineapple silage.

Strain	Halo	
	Salmonella	Listeria	
L. paracasei_ 6709	++	+	
L. paracasei_ 6710	++	++	
L. paracasei_ 6711	++	+	
L. paracasei_ 6712	+++	++	
L. paracasei_ 6713	++	++	
L. paracasei_ 6714	+++	+++	
L. paracasei_ 6715	+	+	
L. fermentum_ 6702	++	+	
L. fermentum_ 6704	+	+	
L. parafarraginis_ 6717	++	++	
L. parafarraginis_ 6719	++	+	
W. ghanensis_ 6706	+++	++	
L. paracasei ATCC 393	+	+	
Notes.

+ Inhibition zone between 0- and 3-mm diameter (weak).

++ Inhibition zone between 3- and 6-mm diameter (good).

+++ Inhibition zone larger than 6-mm diameter (strong).

Table 4 Absorbance values obtained to evaluate the antimicrobial activity of the supernatant of L. paracasei_6714 against Salmonella and L. monocytogenes..

Supernatant volume (µL)	Absorbance at 620 nm	
	Salmonella	L. monocytogenes	
50	0.062 ± 0.007cd	0.043 ± 0.05bc	
45	0.09 ± 0.04cd	0.13 ± 0.02a	
40	0.055 ± 0.008d	0.128 ± 0.004a	
35	0.08 ± 0.03cd	0.14 ± 0.01a	
30	0.15 ± 0.06bcd	0.11 ± 0.05ab	
25	0.16 ± 0.03bcd	0.113 ± 0.004ab	
20	0.19 ± 0.03bc	0.129 ± 0.003a	
15	0.24 ± 0.01ab	0.13 ± 0.01a	
Positive control	0.34 ± 0.08a	0.151 ± 0.007a	
Notes.

Mean values (±standard deviation, n = 3). Values not sharing a common letter represent significantly different values (P <  0.05).

The results for the adhesion to HeLa cells are found in Table 6. According to the cell counts, the adhesion capacity of L. paracasei_6714 was 200% higher than that of L. fermentum (control isolate). The enological capacity of the studied isolate to prevent pathogen invasion is shown in Table 7. In the treatment assay, the adhesion of the pathogen was reduced by approximately 11%. On the other hand, in the protection assay, pathogen reduction was between 10% and 20%.

Discussion

Hostile conditions associated with environmental traits of pineapple peel silages, make the LAB isolated from this matrix, important probiotic or with biotechnological potential. Lactobacilli were the most common group found in this research. These results are similar to other reports of LAB isolated from fermented products (Sáez, Flomenbaum & Zárate, 2018), particularly from pineapple and pineapple waste (Mardalena & Erina, 2016; Arshad et al., 2018). This finding is not surprising due to the exceptional genetic diversity of the Lactobacillus genus, which has recently divided into 23 novel genera (De Bruyne et al., 2010; Di Cagno et al., 2010; Zheng et al., 2020). On the other hand, many Weissella isolates have been obtained from fermentation processes and characterized as heterofermentative bacteria. In fact, W. ghanensis was first isolated from cacao fermentation (De Bruyne et al., 2010).

Figure 2 Cellular auto-aggregation ability of selected lactic acid bacteria (LAB) isolated from pineapple waste and comparison with L. casei ATCC 393.

Data are reported as mean ± SD.

Table 5 Antibiotic resistance/suceptibility of L. paracasei_ 6714.

Antibiotic	Halo (inhibition zone)	Interpretation	
Ciprofloxacin	5.3 (±0,6)	R	
Vancomycin	0.0 (±0)	R	
Penicillin	11.0 (±1.0)	R	
Amoxycilin with clavulanic acid	15.0 (±0,5)	I	
Eritromycin	15.2 (±0,3)	I	
Amikacin	6.0 (±0)	R	
Streptomycin	3.7 (±0,6)	R	
Tetracycline	8.8 (±1)	R	
Chloramphenicol	10.3 (±0,6)	R	
Notes.

Mean values (±standard deviation, n = 3).

R resistant

I intermediate

Table 6 Adhesion of L. paracasei_6714 to HeLa cells per microscopic field.

Strain	LAB adherence to epithelial cells	
L. paracasei_ 6714	403 ± 18	
L. fermentum_ 6702	164 ± 16	

Table 7 Antagonistic effects of L. paracasei_6714 on Salmonella Typhimurium invasion of HeLa cells.

Assays	Log CFU /mL Salmonella	Cell HeLA adhesion (%)	
Treatment	5,3 ± 0,1B	65 ± 1B	
Protection (3 h)a	5,4 ± 0,2B	66 ± 2B	
Protection (24 h)a	4,6 ± 0,1C	56 ± 1C	
Control	6,2 ± 0,1A	76 ± 2A	
Notes.

Mean values (±standard deviation, n = 3). Values not sharing a common letter represent significantly different values (P < 0.05).

a Post-inoculation time with Salmonella Typhimurium.

Isolates were further characterized for their probiotic potential to provide favorable effects on the human gut (Pan et al., 2009). Probiotic evaluation of novel strains must include tolerance to the GI tract, antimicrobial activity, susceptibility to antibiotics, and adhesion to mammalian cells, among others (Byakika et al., 2019). The group of tests for GI tolerance are aimed to evaluate whether the strains are able to survive exposure to acid and enzymes and eventually the transit through the stomach and intestines (Ramos et al., 2013; García-Ruiz et al., 2014; Hernández-Alcántara et al., 2018). In this study, a low tolerance to low pH was observed for most of the isolates, with the exception of L. parafarraginis_6719 which showed the highest survival response (more than 90%). It is important to point out the need to evaluate hundreds of strains to select those that can survive acidic environments (Ramos et al., 2013). However, resistance for all the isolates was higher when compared with the control. It is hypothesized that the tolerance to acidic conditions observed in this study may be related to the ensilage process, in which the LAB that survive the last stages were subjected to acidic pH for a prolonged period of time (Muraro et al., 2021). Besides, these results indicate that some of the isolates may be able to survive the normal gastric environment. It is worth noting that the average pH during human digestion is around 2.0–3.0 with gradients from 1.8 to 4.0 during 2 to 3 h periods (Maragkoudakis et al., 2016). Also, the high survival of LAB to lysozyme exposure in this study was similar to the results previously reported (García-Ruiz et al., 2014) where survival greater than 80% were observed for isolates of L. pentosaceus, L. casei, and L. plantarum after incubation for 120 min; however, survival was around 50% for some isolates. Lysozyme resistance of LAB has been attributed to the peptidoglycan structure in the bacteria cell wall, the physiological state of cells, and the enzyme concentration in the medium (Cunningham, Proctor & Goestsh, 1991; Delfini et al., 2004). The ability to survive in the presence of bile is another important characteristic of potential probiotic strains (García-Ruiz et al., 2014; Hernández-Alcántara et al., 2018). In the case of probiotics, it was established that survival limits for bile salts should be 50% or higher after exposure to a concentration of 0.3% (Mathara et al., 2008). Using these criteria, any of the isolates in this study (after pH and lysozyme tests) were classified as bile-resistant. Still, Bifidobacterium, other Lactobacillus strains, Pediococcus pentosaceus, and some yeasts have been reported as bile resistant according to these criteria (Delgado et al., 2008; Jensen et al., 2012; Turchi et al., 2013; García-Ruiz et al., 2014). To obtain accurate colonization of the host GI tract, a high bile tolerance is a desirable characteristic for bacteria aimed to be used as probiotics (Luo et al., 2012; Byakika et al., 2019). In this research, it was found that bile survival is strain-related instead of LAB species-related and these data are in agreement with previous reports (Delgado et al., 2008; Maldonado et al., 2012).

Inhibitory activity against foodborne pathogens is a desirable trait for bacteria with probiotic potential (Hütt et al., 2006). Previous reports have shown that some LAB strains are able to inhibit both Gram-positive and Gram-negative bacteria by the secretion of organic acids or other antimicrobial compounds such as bacteriocins (Alakomi et al., 2000; Vieco-Saiz et al., 2019). For example, a strong antimicrobial potential was reported for L. acidophilus NIT against Salmonella Typhimurium, Escherichia coli, and Clostridium difficile (Pan et al., 2009). Similar findings were observed from this study as L. paracasei_6714 was active against both Salmonella and L. monocytogenes. A previous report by Hütt et al. (2006) also found an important level of diversity in the antimicrobial activity of different LAB strains, highlighting the importance of an extensive evaluation of newly isolated strains.

The antimicrobial capacity of L. paracasei_6714 in solid media was further corroborated with the supernatant test. Bacterial metabolites in the medium such as lactic acid, acetic acid, diacetyl, and others may be responsible for the observed inhibitory effect (Çon & Gökalp, 2000). Inhibition by L. paracasei_6714 was still observed, even though the supernatant was previously neutralized with NaOH. This suggests that other compounds, such as extracellular proteins as bacteriocins, may be responsible for the observed effect. Several lactobacilli species can excrete antimicrobial proteins (Mora-Villalobos et al., 2020). This property is advantageous in terms of host colonization and competition with other bacteria as other microorganisms are inhibited by the excreted metabolites or through competitive exclusion mechanisms based on competition for binding sites and nutrients (Vieco-Saiz et al., 2019). L. paracasei_6714 is able to synthesize extracellular compounds that can inhibit both Salmonella and L. monocytogenes and it may be able to inhibit pathogens during in vivo applications.

According to García-Cayuela et al. (2014), auto-aggregation is a probiotic property that allows the organism to form cell aggregates which in turn increases the adhesion of cells to the epithelial lining of the intestine and therefore, allowing better colonization of the probiotic organism in the gut. The percentage of auto-aggregation obtained for L. paracasei_6714 after 4 h during this study is greater than 48% (Rastogi, Mittal & Singh, 2020), suggesting a good adhesion capability.

Concerning susceptibility to antibiotics, an important level of resistance was observed for L. paracasei_6714, especially to vancomycin. This antibiotic is considered one of the last resource treatments for multidrug-resistant pathogens, and as a result, this trait is a major concern (Sharma et al., 2016). Previous studies have linked intrinsic resistance to glycopeptides in lactobacilli with the ability to replace the terminal d-alanine residue with d-lactate or d-serine in the muramyl pentapeptide, which prevents vancomycin binding (Sharma et al., 2016). Antibiotic resistance is considered an advantage for probiotic strains as it facilitates the process of host colonization and survival to eventual exposure to antibiotic treatment (Bacha, Mehari & Ashenafi, 2010; Sharma et al., 2014). Nevertheless, there may be a risk of transfer of this feature from antibiotic-resistant strains to foodborne pathogens, since most of the resistance genes are located in gene hotspots along with mobile elements such as plasmids (Oliveira et al., 2017). However, as no plasmids were detected in L. paracasei_6714, the risk for transferring antibiotic resistance traits to other bacteria during in vivo applications should be low.

Finally, the cell culture test was performed to evaluate the ability of L. paracasei_6714 to adhere to intestinal epithelial cells and mucosal surfaces. This is a prerequisite for gut colonization by probiotics (Janković et al., 2012). Colonization and adhesion may be determined by the aggregation of LAB cells (Collado, Meriluoto & Salminen, 2007), which is favored by the formation of a film that contributes to the exclusion of pathogens (Gopal et al., 2001; Tsai et al., 2005). Precisely, L. paracasei_6714 showed a significant level of adhesion to HeLa cells associated with a reduced level of cell infection by Salmonella. Likewise, it was found that LAB reduced cell infection by E. coli by 31% to 52% (García-Ruiz et al., 2014).

Conclusions

Pineapple has been associated with the presence of diverse groups of LAB such as Lactobacillus and Weisella; these bacteria are adapted to the hostile conditions imposed by the nature of this matrix. As in Costa Rica, pineapple production is one of the most important activities within the agro-industrial sector, it might be possible to find an important diversity of strains with potential biotechnological applications in both, the fresh and/or in the by-products derived from the pineapple industry that are used as silage material or are regarded as a waste.

This is the first study analyzing bacteria with potential probiotic features from Costa Rican sources. The results confirm that agro-industrial by-products, specifically silages, may be an important source of promising LAB strains with a potential probiotic and biotechnological profile. At least one of the isolates (L. paracasei_ 6714) obtained could be a potential probiotic candidate based on its in vitro characteristics and behavior. Additional studies, including encapsulation, could improve survival in the GI environment. This isolate showed important antagonistic activity against pathogens of public health concern, antibiotic resistance without the presence of plasmids, and a good adhesion pattern in cell cultures. Further studies to assess its potential use as a beneficial culture in the food industry are highly recommended. Additional tests may include, among others, tolerance to sodium chloride, production of bile salt hydrolase, in vivo tests using animal models, experiments to evaluate the behavior of the isolate in different food matrices, and production of exopolysaccharides.

Supplemental Information

Supplemental Information 1 GenBank accession numbers of 16S rRNA gene and phenylalanyl-tRNA synthase gene (pheS) sequences from lactic acid bacteria (LAB) isolated from pineapple peel silage

Click here for additional data file.

Supplemental Information 2 Picture of plaques and the observed inhibition halo of L. paracasei 6712 and L. paracasei_6714 against L. monocytogenes (A, B) and Salmonella sp. (C, D)

Click here for additional data file.

Supplemental Information 3 Picture of gel red stained agarose gel (0.8%) electrophoresis

Gel order:100 bp MassRuler DNA ladder, miniprep of L. paracasei_6714, and miniprep of positive control.

Click here for additional data file.

Supplemental Information 4 GenBank sequences from this research

Click here for additional data file.

Supplemental Information 5 Fasta sequences for 16S data (raw data)

Click here for additional data file.

Supplemental Information 6 Aligment of fasta sequences for 16S data (raw data)

Click here for additional data file.

Supplemental Information 7 Fasta sequences for pheS data (raw data)

Click here for additional data file.

Supplemental Information 8 Aligment of fasta sequences for pheS data (raw data)

Click here for additional data file.

Supplemental Information 9 Raw data for ph2, bile and lysozyme

Click here for additional data file.

Supplemental Information 10 Antagonist and Antimicrobial activity

Click here for additional data file.

Supplemental Information 11 Antibiotic resistance (Table 5, raw data)

Click here for additional data file.

Supplemental Information 12 Cell culture raw data

Click here for additional data file.

This work was supported by the technical assistance of Henry Castro, Arturo Pacheco, Vanny Mora, and María Fernanda Miranda at the Food Microbiology Laboratory and Molecular Biology Laboratory from UCR. Also, the authors acknowledge M.Sc. Marlen Cordero Serrano from CIET for her technical support during the cell culture assays.

Additional Information and Declarations

Competing Interests

Author Contributions

DNA Deposition

Data Availability

The authors declare there are no competing interests.

Jannette Wen Fang Wu Wu conceived and designed the experiments, performed the experiments, analyzed the data, prepared figures and/or tables, authored or reviewed drafts of the paper, and approved the final draft.

Mauricio Redondo-Solano and Jessie Usaga performed the experiments, analyzed the data, prepared figures and/or tables, authored or reviewed drafts of the paper, and approved the final draft.

Lidieth Uribe and Rodolfo WingChing-Jones analyzed the data, authored or reviewed drafts of the paper, and approved the final draft.

Natalia Barboza conceived and designed the experiments, performed the experiments, analyzed the data, authored or reviewed drafts of the paper, and approved the final draft.

The following information was supplied regarding the deposition of DNA sequences:

All accessions are kept with the same name indicated on this research in the Bacteriology Collection at the Faculty of Microbiology and in the Bacteriology Collection at the National Center for Food Science and Technology (CITA), University of Costa Rica.

Lactobacillus casei strain Lc-P6709 16S ribosomal RNA gene, partial sequence

GenBank: MH753098.1

https://www.ncbi.nlm.nih.gov/nuccore/MH753098.1?report=fasta

Lactobacillus paracasei strain Lp-P6710 16S ribosomal RNA gene, partial sequence

GenBank: MH753094.1

https://www.ncbi.nlm.nih.gov/nuccore/MH753094.1?report=fasta

Lactobacillus paracasei strain Lp-P6711 16S ribosomal RNA gene, partial sequence

GenBank: MH753095.1

https://www.ncbi.nlm.nih.gov/nuccore/MH753095.1?report=fasta

Lactobacillus paracasei strain Lp-P6712 16S ribosomal RNA gene, partial sequence

GenBank: MH753096.1

https://www.ncbi.nlm.nih.gov/nuccore/MH753096.1?report=fasta

Lactobacillus casei strain Lc-P6713 16S ribosomal RNA gene, partial sequence

GenBank: MH753099.1

https://www.ncbi.nlm.nih.gov/nuccore/MH753099.1?report=fasta

Lactobacillus paracasei strain Lp-P6714 16S ribosomal RNA gene, partial sequence

GenBank: MH753097.1

https://www.ncbi.nlm.nih.gov/nuccore/MH753097.1?report=fasta

Lactobacillus casei strain Lc-P6715 16S ribosomal RNA gene, partial sequence

GenBank: MH753100.1

https://www.ncbi.nlm.nih.gov/nuccore/MH753100.1?report=fasta

Lactobacillus fermentum strain Lf-P6702 16S ribosomal RNA gene, partial sequence

GenBank: MH753090.1

https://www.ncbi.nlm.nih.gov/nuccore/MH753090.1?report=fasta

Lactobacillus fermentum strain Lf-P6704 16S ribosomal RNA gene, partial sequence

GenBank: MH753091.1

https://www.ncbi.nlm.nih.gov/nuccore/MH753091.1?report=fasta

Lactobacillus parafarraginis strain Lp-P6717 16S ribosomal RNA gene, partial sequence

GenBank: MH753092.1

https://www.ncbi.nlm.nih.gov/nuccore/MH753092.1?report=fasta

Lactobacillus parafarraginis strain Lp-P6719 16S ribosomal RNA gene, partial sequence

GenBank: MH753093.1

https://www.ncbi.nlm.nih.gov/nuccore/MH753093.1?report=fasta

Weissella ghanensis strain Wg-P6706 16S ribosomal RNA gene, partial sequence

GenBank: MH753101.1

https://www.ncbi.nlm.nih.gov/nuccore/MH753101.1?report=fasta

Lactobacillus casei strain P6709 phenylalanine-tRNA ligase subunit alpha (pheS) gene, partial cds

GenBank: MH752084.1

https://www.ncbi.nlm.nih.gov/nuccore/MH752084.1?report=fasta

Lactobacillus paracasei strain P6710 phenylalanine-tRNA ligase subunit alpha (pheS) gene, partial cds

GenBank: MH752080.1

https://www.ncbi.nlm.nih.gov/nuccore/MH752080.1?report=fasta

Lactobacillus paracasei strain P6711 phenylalanine-tRNA ligase subunit alpha (pheS) gene, partial cds

GenBank: MH752081.1

https://www.ncbi.nlm.nih.gov/nuccore/MH752081.1?report=fasta

Lactobacillus paracasei strain P6712 phenylalanine-tRNA ligase subunit alpha (pheS) gene, partial cds

GenBank: MH752082.1

https://www.ncbi.nlm.nih.gov/nuccore/MH752082.1?report=fasta

Lactobacillus casei strain P6713 phenylalanine-tRNA ligase subunit alpha (pheS) gene, partial cds

GenBank: MH752085.1

https://www.ncbi.nlm.nih.gov/nuccore/MH752085.1?report=fasta

Lactobacillus paracasei strain P6714 phenylalanine-tRNA ligase subunit alpha (pheS) gene, partial cds

GenBank: MH752083.1

https://www.ncbi.nlm.nih.gov/nuccore/MH752083.1?report=fasta

Lactobacillus casei strain P6715 phenylalanine-tRNA ligase subunit alpha (pheS) gene, partial cds

GenBank: MH752086.1

https://www.ncbi.nlm.nih.gov/nuccore/MH752086.1?report=fasta

Lactobacillus fermentum strain P6702 phenylalanine-tRNA ligase subunit alpha (pheS) gene, partial cds

GenBank: MH752076.1

https://www.ncbi.nlm.nih.gov/nuccore/MH752076.1?report=fasta

Lactobacillus fermentum strain P6704 phenylalanine-tRNA ligase subunit alpha (pheS) gene, partial cds

GenBank: MH752077.1

https://www.ncbi.nlm.nih.gov/nuccore/MH752077.1?report=fasta

Lactobacillus parafarraginis strain P6717 phenylalanine-tRNA ligase subunit alpha (pheS) gene, partial cds

GenBank: MH752078.1

https://www.ncbi.nlm.nih.gov/nuccore/MH752078.1?report=fasta

Lactobacillus parafarraginis strain P6719 phenylalanine-tRNA ligase subunit alpha (pheS) gene, partial cds

GenBank: MH752079.1

https://www.ncbi.nlm.nih.gov/nuccore/MH752079.1?report=fasta

The following information was supplied regarding data availability:

The sequences (16S, 1, pheS), assays of resistance to the gastrointestinal tract (pH2, bile, lisozyme), cell culture, antibiotics, antagonist and antimicrobial activity, and plasmid DNA isolation (agarose gel) are available in the Supplemental Files.

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
