# Peer review of "First characterization of the probiotic potential of lactic acid bacteria isolated from Costa Rican pineapple silages"

_PeerJ, doi:10.7717/peerj.12437_

## Round 0.1 · original submission · Major Revisions

There are critical points that need to be included in the manuscript. Please revise the concerns and comments made by reviewers.

Revision of this work does not guarantee the potential acceptance.

·

Basic reporting

The work is novel and relevant.

Experimental design

The study was well desgined.

Validity of the findings

The data are convincing and the conclusion justified.

Additional comments

The present study described the isolation of several lactic acid bacteria (LAB) from Costa Rican pineapple silage and characterization of their probiotic potential. As a result, an LAB strain, Lacticaseibacillus paracasei_6714, was identified as a promising probiotic candidate. The work was well-designed and presented.

Major concerns:
1. The antimicrobial activity needs to be evaluated using more LAB strains of different species.
2. The safety of the LAB isolates needs to be tested.
3. In vivo experiments are suggested to evaluate the probiotic ability of the LAB.
4. Well-identified reference LAB strains need to be included as controls in all the assays.
5. The novelty and innovation of the study should be highlighted.
6. The manuscript should be revised by native English speakers.

Minor points:
1. The layout of the tables and figures should be optimized.
2. The references should be a consistent style.
3. Line 1, lactic acid bacteria
4. Line 33, species
5. Lines 144-145, A 1.5- kb fragment…..the primer pair 27F/1492R
6. Line 273, a 1-kb
7. Lines 278 and 295, HeLa
8. Lines 327-328, L. paracasei
9. Line 331, GenBank
10. Line 397, 0.3%

Reviewer 2 ·

Basic reporting

See notes to authors

Experimental design

ok

Validity of the findings

ok

Additional comments

Line 60: use the WHO/FAO 2022 or ISAPP 2014 definition of probiotics, and cite either source.
Line 63: to colonize the gut is not a prerequisite for probiotics, was not even demonstrated for any strains. Colonization is only temporary, please comment.
Line 101: this is not true, many more studies were conducted in southamerica, especially in Brazil by Prof. R. Schwan https://pubmed.ncbi.nlm.nih.gov/?term=Schwan+RF&cauthor_id=32869919
Line 108: this is not true either, many other were conducted in brazil with LAB from coca: https://pubmed.ncbi.nlm.nih.gov/?term=brazil+cocoa+lactic+acid+bacteria please update.
The introduction is too long, almost 3 pages, it should be reduced by half and transfer the remaining text to discussion.
How many samples were used for isolation? How many colonies were obtained from each sample for further identification? How many isolates failed to be identified as LAB? What other species non LAB were found?
Line 328: was RAPD applied to say that the strains were really different strains? Most isolates from the same sample are indeed same strains, so it is important to indicate in a table if isolates were obtained from the same sample, or if RAPD was done to make sure that the seven strains are really 7 different strains or copies of the same isolate.
Line 453: this sentence: “and one of few studies obtaining LAB with biotechnological potential from agro-industrial waste in Latin America” shows a limited knowledge of what was published, please deleted or update it.
Tables: do not express results of survival as %, show the colony counts before and after exposure ot acidity, lysozyme, bile salts, etc.

---

## Round 0.2 · Minor Revisions

Please revise according to the comments made by the reviewer.

Reviewer 2 ·

Basic reporting

Most issues were addressed

Experimental design

ok

Validity of the findings

ok

Additional comments

Still authors must consider to refer to isolates, not to strains, unless they show by RAPD that isolates are indeed different strains.

In particular in this paragraph:

A total of twelve different LAB morphotypes were isolated from twenty pineapple silages with
334 increasing levels of urea. Considering the 16S rRNA sequence and pheS gen the isolates
335 correspond to L. paracasei (seven strains), Lentilactobacillus parafarraginis (two strains),
336 Limosilactobacillus fermentum (two strains), and W. ghanensis (one strain)

---

## Round 0.3 · accepted · Accept

All comments and suggestions raised by reviewers have been covered by the authors.